# Severity of bovine tuberculosis is associated with innate immune-biased transcriptional signatures of whole blood in early weeks after experimental *Mycobacterium bovis* infection

Jayne E. Wiarda [1,2,3], Paola M. Boggiatto [1]*, Darrell O. Bayles[1], W. Ray Waters[1], Tyler C. Thacker [1], Mitchell V. Palmer[1]

1 Infectious Bacterial Diseases Research Unit, National Animal Disease Center, Agricultural Research Service, United States Department of Agriculture, Ames, IA, United States of America, 2 Immunobiology Graduate Program, Iowa State University, Ames, IA, United States of America, 3 Oak Ridge Institute for Science and Education, Agricultural Research Service Participation Program, Oak Ridge, TN, United States of America

* paola.boggiatto@usda.gov

**Data Availability Statement:** Sequence fastq files used for pipeline analysis of RNA-seq data can be

## Abstract

*Mycobacterium bovis*, the causative agent of bovine tuberculosis, is a pathogen that impacts both animal and human health. Consequently, there is a need to improve understanding of disease dynamics, identification of infected animals, and characterization of the basis of immune protection. This study assessed the transcriptional changes occurring in cattle during the early weeks following a *M. bovis* infection. RNA-seq analysis of whole blood-cell transcriptomes revealed two distinct transcriptional clusters of infected cattle at both 4- and 10-weeks post-infection that correlated with disease severity. Cattle exhibiting more severe disease were transcriptionally divergent from uninfected animals. At 4-weeks post-infection, 25 genes had commonly increased expression in infected cattle compared to uninfected cattle regardless of disease severity. Ten weeks post-infection, differential gene expression was only observed when severely-affected cattle were compared to uninfected cattle. This indicates a transcriptional divergence based on clinical status following infection. In cattle with more severe disease, biological processes and cell type enrichment analyses revealed overrepresentation of innate immune-related processes and cell types in infected animals. Collectively, our findings demonstrate two distinct transcriptional profiles occur in cattle following *M. bovis* infection, which correlate to clinical status.

## Introduction

*Mycobacterium bovis* is a member of the *Mycobacterium tuberculosis* complex that preferentially infects cattle (*Bos taurus*) and is the primary causative agent of bovine tuberculosis (bTB). Though most cases of human tuberculosis (hTB) are attributed to *Mycobacterium tuberculosis* infection, *M. bovis* can also cause infection in humans, known as zoonotic TB. Eradication programs in the United States, primarily based on test and slaughter, have

found at SRA database (accession number PRJNA600004). RNA-seq pipeline bash and R scripts as well as output and input data for DGE analyses can be obtained at https://github.com/jwiarda/EarlyTB_RNAseq. The remainder relevant data are within the manuscript and its Supporting Information files.

**Funding:** Research was funded by intramural funds from the United States Department of Agriculture, Agricultural Research Service Project, CRIS #5030-32000-222.

**Competing interests:** The authors have declared that no competing interests exist.

generally had great success in controlling bTB, but total eradication has not been achieved and remains unlikely [1, 2]. A variety of factors including the movement of infected livestock [3], the presence of wildlife reservoirs [1, 3], the cost of managing infected cattle [1], and the lack of effective vaccines [4, 5] have contributed to the continued disease presence in both livestock and wildlife species. In 1995, it was estimated that worldwide, >50 million cattle were infected with *M. bovis*, resulting in a financial loss of $3 billion USD annually [6]. Despite efforts to reduce bTB incidence worldwide, disease caused by *M. bovis* remains a significant economic burden and has a major impact on animal and human health. Improving our understanding of disease dynamics, identifying infected animals, and characterizing the basis of immune protection are key to reducing the incidence and severity of disease caused by *M. bovis*.

Transcriptome analyses can provide insights that help us understand complex diseases including the underlying mechanisms of pathogenesis, and the genes responsible for protective immune responses. Genes associated with specific diseases, known as biomarkers, once identified, can be used in disease diagnosis, determination of clinical status, and assessment of disease progression. Previous bTB studies employing microarrays or RNA sequencing (RNA-seq) have identified over 3,000 differentially expressed genes in cattle naturally-infected with *M. bovis* compared to uninfected cattle, and pathway analyses revealed enrichment for genes involved with immune function [7–11].

However, analysis of transcriptional dynamics of bTB at early post-infection timepoints is lacking. In cattle, pulmonary lesions associated with experimental *M. bovis* infection via aerosol have been noted as early as 15 days after infection [12] and robust interferon-gamma (IFN-γ) responses to both complex and specific *M. bovis* antigens are seen 3 to 4 weeks after infection [13–15]. Studies using a non-human primate model of human *M. tuberculosis* infection have clearly demonstrated that early post-exposure interactions between pathogen and host are critical in determining the eventual outcome of infection [16–18].

To gain a deeper knowledge into the transcriptional dynamics occurring in the early weeks after *M. bovis* infection, we performed differential gene expression analysis of RNA-seq data from whole blood leukocyte populations collected with a cell filter-based system in cattle experimentally-infected with aerosolized *M. bovis*. In this manuscript, we present gene expression profiles, pathway and cell type enrichment analysis, and correlate our transcriptional findings to lesion severity and clinical disease presentation.

## Materials and methods

### Animals

Holstein steers (n = 13) of 2–3 months of age were obtained from Kline Dairy (State Center, Iowa), a source with no history of bTB. Cattle were housed outdoors for ~1.5 months, then transitioned to a biosafety level 3 (BSL-3) indoor facility where they were randomly assigned to 2 rooms corresponding to treatment: infected (n = 8) or uninfected (n = 5). After ~2 additional months of acclimation to BSL-3 housing, cattle in the infected group (~5–6 months of age) were experimentally infected with *M. bovis*. Cattle were monitored daily for development of clinical signs (coughing, labored breathing, fever).

Calves were humanely euthanized by intravenous administration of sodium pentobarbital at 6 (n = 1), 10 (n = 2), and 12 months (n = 3) after infection. Animals displaying all three clinical signs (coughing, labored breathing, and a body temperature >39.3˚C) (n = 2) concurrently, were euthanized at 2.5–3 months post-infection, within 24 hours of meeting the criteria. No animals died before meeting the criteria for humane endpoint euthanasia.

All experimental animal procedures were conducted in accordance with recommendations in the Care and Use of Laboratory Cattle of the National Institutes of Health and the Guide for

the Care and Use of Agricultural Cattle in Research and Teaching [19, 20]. All procedures performed were approved by the USDA-National Animal Disease Center Animal Care and Use Committee, under procolol number ARS-2015-451.

## Inoculum preparation and aerosol challenge

*Mycobacterium bovis* strain 10–7428 was used in this experiment [21]. This field strain was of low passage (less than 3 passages) and has been previously shown to be virulent in the calf aerosol model [22]. Inoculum was prepared using standard techniques [23] in Middlebrook's 7H9 liquid media (Becton Dickinson, Franklin Lakes, NJ, USA) supplemented with 10% oleic acid-albumin-dextrose complex (OADC; Difco, Detroit, MI, USA) plus 0.05% Tween 80 (Sigma Chemical Co., St. Louis, MO, USA). Mid log-phase growth bacilli were pelleted by centrifugation at 750 x g, washed twice with phosphate buffered saline (PBS) (0.01 M, pH 7.2) and stored at -80°C until used. Frozen stock was warmed to room temperature and diluted to the appropriate cell density in 2 ml of 7H9 liquid media. Bacilli were enumerated by serial dilution plate counting on Middlebrook's 7H11 selective media (Becton Dickinson). A single dose was determined to be $1.77 \times 10^6$ CFU per calf.

Aerosol infection of calves with virulent *M. bovis* has been described in detail previously [22, 24, 25]. Briefly, 8 Holstein steer calves (5–6 months of age) were infected with a single dose of virulent *M. bovis* strain 10–7428 by nebulization of inoculum into a mask (Equine AeroMask®, Trudell Medical International, London, ON, Canada) covering the nostrils and mouth.

## Post-mortem examination

Following euthanasia, tissues were examined for gross lesions and processed for microscopic analysis as described previously [22]. Tissues collected included: lung, liver, and, lymph nodes (medial retropharyngeal, mediastinal, tracheobronchial, hepatic, and mesenteric). Grossly, lymph nodes and lung lobes were examined on cut-surface in 0.5–1 cm sections. Lungs and lymph nodes were scored using a semiquantitative gross lesion scoring system [26]. Lung lobes (left cranial, left caudal, right cranial, right caudal, middle and accessory) were assessed individually based on the following scoring system: 0, no visible lesions; 1, no external gross lesions but lesions seen on slicing; 2, < 5 gross lesions of < 10 mm in diameter; 3, > 5 gross lesions of < 10 mm in diameter; 4, > 1 distinct gross lesion of > 10 mm in diameter; and 5, gross coalescing lesions. Cumulative mean scores were then calculated for each entire lung. Lung-associated lymph nodes (tracheobronchial and mediastinal) were weighed and scored. Scoring of lymph node pathology was based on the following system: 0, no necrosis or visible lesions; 1, small focus (1–2 mm diameter); 2, several small foci; 3, extensive necrosis.

Tissues collected for microscopic analysis were fixed in 10% neutral buffered formalin (3.7% formaldehyde) for ~24 hours, transferred to 70% ethanol and processed routinely. Formalin-fixed, paraffin embedded sections (4 μm) were stained with hematoxylin and eosin (H&E). Adjacent sections from samples containing lesions consistent with tuberculosis were stained by the Ziehl-Neelsen technique for identification of acid-fast bacteria (AFB).

The same tissues collected for microscopic analysis were collected for mycobacterial isolation as described previously [27] using Middlebrook 7H11 selective agar plates (Becton Dickinson) incubated for 8 weeks at 37°C. Colonies were confirmed as *M. bovis* using IS6110 real time PCR as described previously [28].

## Leukocyte RNA sample collection

Total leukocyte populations were isolated from whole blood using the LeukoLOCK™ Fractionation & Stabilization Kit (Ambion, Life Technologies, Carlsbad, CA, USA) according to

manufacturer's instructions (Life Technologies). Briefly, 9 to 10 mL of whole blood were collected from each animal via jugular venipuncture into a $K_2$ EDTA vacutainer tube (Becton, Dickinson and Company, Franklin Lakes, NJ, USA). Blood was immediately filtered through a LeukoLOCK$^{TM}$ Filter apparatus to retain total leukocyte populations while excluding red blood cells. Filters were then rinsed with 3 mL of PBS and flushed with 3 mL of RNA*later*$^{TM}$ (Ambion) to stabilize cells on the filter. Filters containing leukocytes were left saturated in RNA*later*$^{TM}$ and stored at -80˚C up to 25 months before isolating RNA.

## RNA isolation

Isolation of total RNA directly from stored leukocyte filters was performed according to manufacturer's instructions for the alternative protocol for extraction of RNA from cells captured on LeukoLOCK$^{TM}$ Filters using TRI Reagent® (Life Technologies). Briefly, LeukoLOCK$^{TM}$ Filters were thawed to room temperature, flushed using an empty retracted syringe to expel RNA*later*$^{TM}$, with 4 mL TRI Reagent® Solution (Ambion) to lyse cells, collecting the lysate. As a phase separation reagent, 800 µL bromo-3-chloro-propane (BCP) (Acros Organics, Thermo Fisher Scientific, Waltham, MA, USA) was added to the expelled lysate. The lysate/BCP mixture was shaken vigorously for 30 sec, incubated at room temperature for 5 min, and centrifuged at 2,000 x g for 10 min to allow separation of the aqueous and organic phases. The upper aqueous phase containing the RNA was collected, and 0.5 volumes of nuclease-free water and 1.25 volumes ACS reagent grade 100% ethanol (Sigma-Aldrich, St. Louis, MO, USA) were added to the extracted aqueous phase and mixed thoroughly to recover total RNA. The mixture was then filtered through a silica Filter Cartridge (Ambion) using brief centrifugations, and flow-through was discarded. Filters containing RNA were next washed once with 750 µL Wash 1 (a 3:7 ratio of Denaturing Lysis Solution (Ambion) and 100% ethanol, respectively) and twice with 750 µL Wash 2 (an 80:19:1 ratio of 100% ethanol, nuclease-free water, and 5 M NaCl (Ambion), respectively), discarding flow-through after each brief centrifugation. As a stabilization reagent, 250 µL 0.1 mM ethylenediaminetetraacetic acid (EDTA) (Ambion) warmed to 80˚C was added to the center of the filter and incubated at room temperature for 1 min. The filter was centrifuged at max speed for 1 min, and the flow-through containing the eluted RNA was retained.

The RNA elution was treated with DNase from the DNA-free$^{TM}$ Kit (Ambion) according to manufacturer's instructions (Life Technologies). To deplete DNA, 0.1 volumes 10X DNase I Buffer and 2 µL rDNase were added to the RNA elution and incubated at 37˚C for 30 min. Next, 0.1 volumes resuspended DNase Inactivation Reagent were added and incubated at room temperature for 2 min. Tubes were centrifuged at 10,000 x g for 1.5 min, and the supernatant containing the DNase-treated RNA was collected. Samples were stored at -80˚C.

## RNA quality assessment and quantification

RNA quality was assessed via the Agilent RNA 6000 Nano Kit (Agilent Technologies, Santa Clara, CA, USA) run on an Agilent 2100 Bioanalyzer System (Agilent Technologies) according to manufacturer's instructions (Agilent Technologies) and assessed for purity and quantity via NanoDrop$^{TM}$ 2000 Spectrophotometer (Thermo Fisher Scientific).

## RNA-seq library preparation, sequencing, and primary analysis

Isolated total RNA was submitted to the Iowa State University DNA Facility for library preparation, sequencing, and primary analysis. Sample libraries were prepared using the Universal Plus mRNA-Seq kit (NuGEN Technologies, San Carlos, CA, USA) according to the manufacturer's instructions (NuGEN Technologies) and were sequenced as 2x100 paired end, stranded

mRNA-seq libraries using an Illumina HiSeq 3000 (Illumina, San Diego, CA, USA). Sequencing lanes and multiplexing barcodes were randomly assigned to samples to avoid confounding nuisance factors (sequencing lane and barcode assignment) with treatment variables (individual animal, infection group, collection timepoint). Image analysis and sequence base calling were performed, and samples were demultiplexed to obtain processed raw data in fastq file format for both forward and reverse strands. Samples were sequenced to an average raw depth of 25,259,821± 6,231,306 SD, an adequate sequencing depth based on previous findings [8, 29–32].

## RNA-seq pipeline

Read quality was assessed using FastQC (version 0.11.5) [33]. Following quality analysis, reads were trimmed to remove low quality and adapter sequences using Trimmomatic (version 0.36) [34]. FastQC analysis of trimmed-read quality analysis indicated that further read filtering and trimming was not required. After trimming, an average of 20.5 M ± 5.5 M SD paired reads remained in each sample. Paired-end reads were aligned to the *Bos taurus* reference genome (*B. taurus* UMD 3.1.1) [35] obtained from Ensembl [36] using the STAR RNA-seq aligner package (version 2.5.3a) [37]. On average, 92.47% of paired-end reads were uniquely aligned, and 3.96% of read pairs mapped to multiple loci. The number of aligned reads mapping to each gene annotated in the *B. taurus* genome was determined for each library using featureCounts from the Subread package (version 3.22.0) [38]. On average, 82.9% of aligned read pairs could be assigned to an annotation feature, resulting in an average of 18.0 M ± 4.6 M SD assigned read pairs per sample.

## Differential gene expression analysis

Analysis of differential gene expression (DGE) was performed using the gene-wise count tables and the edgeR Bioconductor package (version 3.24.3) [39] based on a negative binomial model. The edgeR package was used to: 1) filter out genes with under 1 count per million (cpm) reads in a minimum threshold of samples; 2) obtain a normalization factor (the relative sequencing depth of each library using the trimmed mean of *M*-values (TMM) method of normalization) [40]; 3) create a design matrix using group-means parameterization to obtain a coefficient for expression to describe each treatment group; 4) estimate the dispersion parameter, $\phi_g$, used to estimate variances by squeezing gene-specific tagwise dispersions towards trended dispersions using empirical Bayes estimations under the Cox-Reid adjusted profile likelihood method [41]; 5) fit a generalized linear model (GLM) [41] to the non-normally distributed data based on the design matrix and $\phi_g$; 6) test for differential expression using the likelihood ratio test (LRT) [41], specifying contrasts for the comparisons of interest; and 7) control false discovery rate (FDR) using the Benjamini-Hochberg method [42] to account for multiple testing error.

Comparisons were made between all infected, clusters of infected, and uninfected cattle at single timepoints. Genes with an FDR value (adjusted p-value) of less than 0.05 were considered to be differentially expressed.

## Biological process enrichment analysis

Lists of genes with increased ($\log_2$FC > 0 and FDR < 0.05) or decreased ($\log_2$FC < 0 and FDR < 0.05) expression obtained through differential gene expression analysis were individually analyzed using ClueGO (version 2.5.5) in Cytoscape (Cytoscape Consortium) [43, 44]. Enriched GO biological processes within the *B. taurus* gene database were detected with the following parameters: medium network specificity, GO term fusion enabled, GO tree interval

between 5 and 20, a minimum of 2 genes and 3% of genes, a kappa score of 0.4, and right-sided hypergeometric testing for enrichment. The list of reference genes was further reduced by only considering genes analyzed during differential gene expression analysis after filtering out of low count genes. Significance values were calculated using Benjamini-Hochberg correction. Corrected p-values < 0.05 were considered significant. Enriched GO immune system processes were detected using previous parameters with the following modifications: GO tree interval between 2 and 20, a minimum of 1.5% of genes, and a p-value < 0.1.

### Cell type enrichment analysis

Cell type enrichment analysis was completed using Cten [45] by inputting lists of genes with increased ($\log_2 FC > 0$ and FDR < 0.05) and decreased ($\log_2 FC < 0$ and FDR < 0.05) expression obtained through differential gene expression analysis. Genes were compared to mouse gene symbols in order to obtain lists of enriched cell types. Significance values were obtained from -$\log_{10}$ Benjamini-Hochberg adjusted p-values. Enrichment scores greater than 2 were considered significant. A heatmap of enrichment scores of cell types showing significant enrichment in at least one comparison was created using GraphPad Prism 8.3.0 for Mac OSX, (GraphPad Software, San Diego, CA, USA).

### Data availability

Sequence fastq files used for pipeline analysis of RNA-seq data can be found at SRA database (accession number PRJNA600004). RNA-seq pipeline bash and R scripts as well as output and input data for DGE analyses can be obtained at https://github.com/jwiarda/EarlyTB_RNAseq.

## Results

### Study overview

The purpose of this study was to assess transcriptional changes during the early stages of *M. bovis* infection in cattle and how these corresponded to disease outcome, as indicated by clinical signs of disease and post-mortem pathology (Fig 1). RNA from whole blood of experimenally-infected cattle and age-matched uninfected cohorts was collected prior to infection (0 wpi), and at 4 and 10 weeks-post infection (wpi) using a filter-based system [46, 47]. Concurrently, clinical signs of disease were monitored, and cattle were euthanized as disease became evident. Upon euthanasia, post-mortem analysis was performed to assess severity of disease pathology.

RNA was later isolated from stored filters using methods shown to yield high quality RNA for high-throughput analyses [48, 49] and analyzed for integrity, purity, and quantity. All samples displayed RINs $\geq$ 7.9 (mean RIN score of 9.09 ± 0.49 SD) and 260/280 ratios ranging from 1.98 to 2.11 (mean absorbance ratio of 2.04 ± 0.03 SD), characteristic of high RNA integrity and purity. Total RNA concentrations ranged between 65.05 and 177.35 ng/μL (mean concentration of 108.41 ± 29.74 ng/μL SD), sufficient for downstream sequencing and analysis. High-quality RNA was used to create cDNA libraries, perform RNA-seq, and analyze data for differential gene expression, biological process enrichment, and cell type enrichment.

### Transcriptionally-distinct clusters following infection

A multidimensional scaling (MDS) plot was used to visualize the transcriptional relationship between samples and observe differences between samples based on infection status and timepoint post-infection (Fig 2A). Gene counts from each timepoint were also analyzed to obtain lists of differentially expressed genes (FDR < 0.05) between infected and uninfected cattle. Of

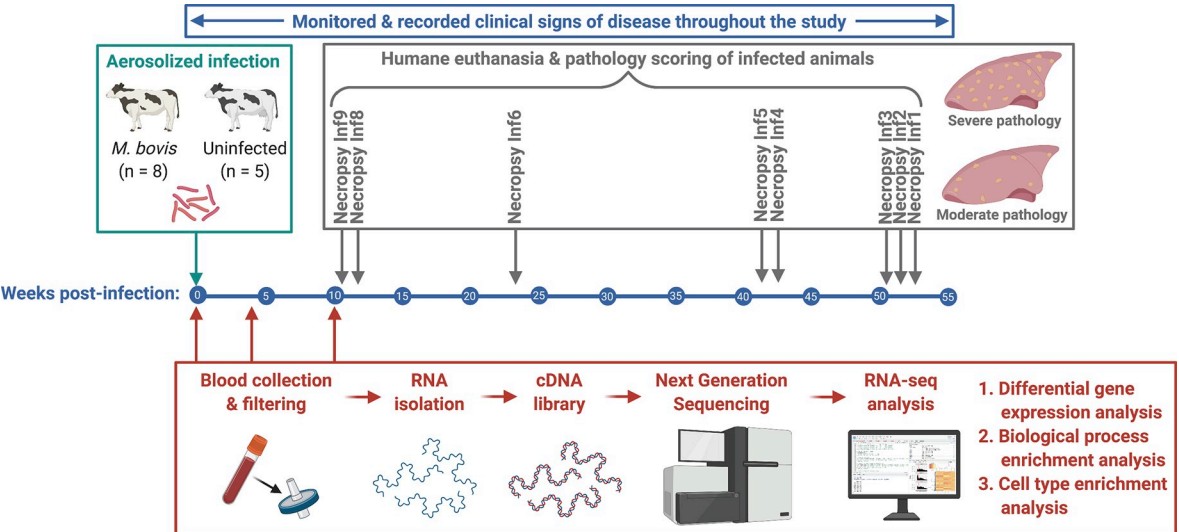

**Fig 1. Overview of experimental timeline and workflow.** Cattle were infected with *M. bovis* (n = 8) or left uninfected (n = 5). Following infection, whole blood was collected, clinical signs of disease were monitored, and pathology grading was assessed at necropsies. Whole blood was collected, and RNA was stored on filters from *M. bovis*-infected (n = 8) and uninfected (n = 5) cattle at 0 wpi, 4 wpi, and 10 wpi. RNA was later isolated and processed to create cDNA libraries that were sequenced, and RNA-seq data were analyzed for differential gene expression, biological process enrichment, and cell type enrichment. Clinical signs of disease were assessed throughout the study, and animals were euthanized as signs of disease became evident. Immediately following euthanasia, necropsies were performed to assess severity of post-mortem pathology.

the 14,279 analyzed genes, a total of 1 (0.01%), 1060 (7.42%), and 162 (1.13%) genes were differentially expressed between infected and uninfected cattle at 0 wpi, 4 wpi, and 10 wpi, respectively (Fig 2B–2D and S1 File).

Clustering of samples based on differentially expressed genes revealed the existence of two distinct groups of infected cattle, observed at both 4 wpi and 10 wpi (Fig 2E and 2F). This clustering was not present at 0 wpi, suggesting that these changes in gene expression were driven by infection (Fig 2G–2I). While one cluster shared more transcriptional similarities with uninfected samples, the other appeared more distantly related to the uninfected samples. Interestingly, samples from one animal, Inf1, clustered to the more distantly related cluster at 4 wpi but with the more closely related cluster at 10 wpi. All other samples corresponding to one animal were found within the same cluster at both timepoints.

## Correlation between disease severity and transcriptional clustering

We evaluated whether the divergence in gene expression profiles of infected animals correlated to clinical presentation and/or disease-associated pathology by assessing disease severity via clinical and post-mortem parameters. Of the 8 infected cattle, 4 developed clinical signs characterized by intermittent coughing (Table 1). Additionally, 2 of these calves became febrile (> 39.3˚C) and clinical signs progressed to labored breathing. These animals were then humanely euthanized at approximately 2.5–3 months post-infection. The 6 remaining calves were euthanized and examined at approximately 6 months (n = 1), 10 months (n = 2) and 12 months (n = 3) post-infection. Interestingly, the 4 animals exhibiting clinical signs were the same animals clustering more distantly from uninfected cohorts (Fig 2E and 2F).

Post-mortem, gross and microscopic lesions (caseonecrotic granulomas with AFB) of variable severity were present in pulmonary lymph nodes (mediastinal and tracheobronchial) in all lung lobes from all experimentally-infected calves (Fig 3A and 3B and Table 2).

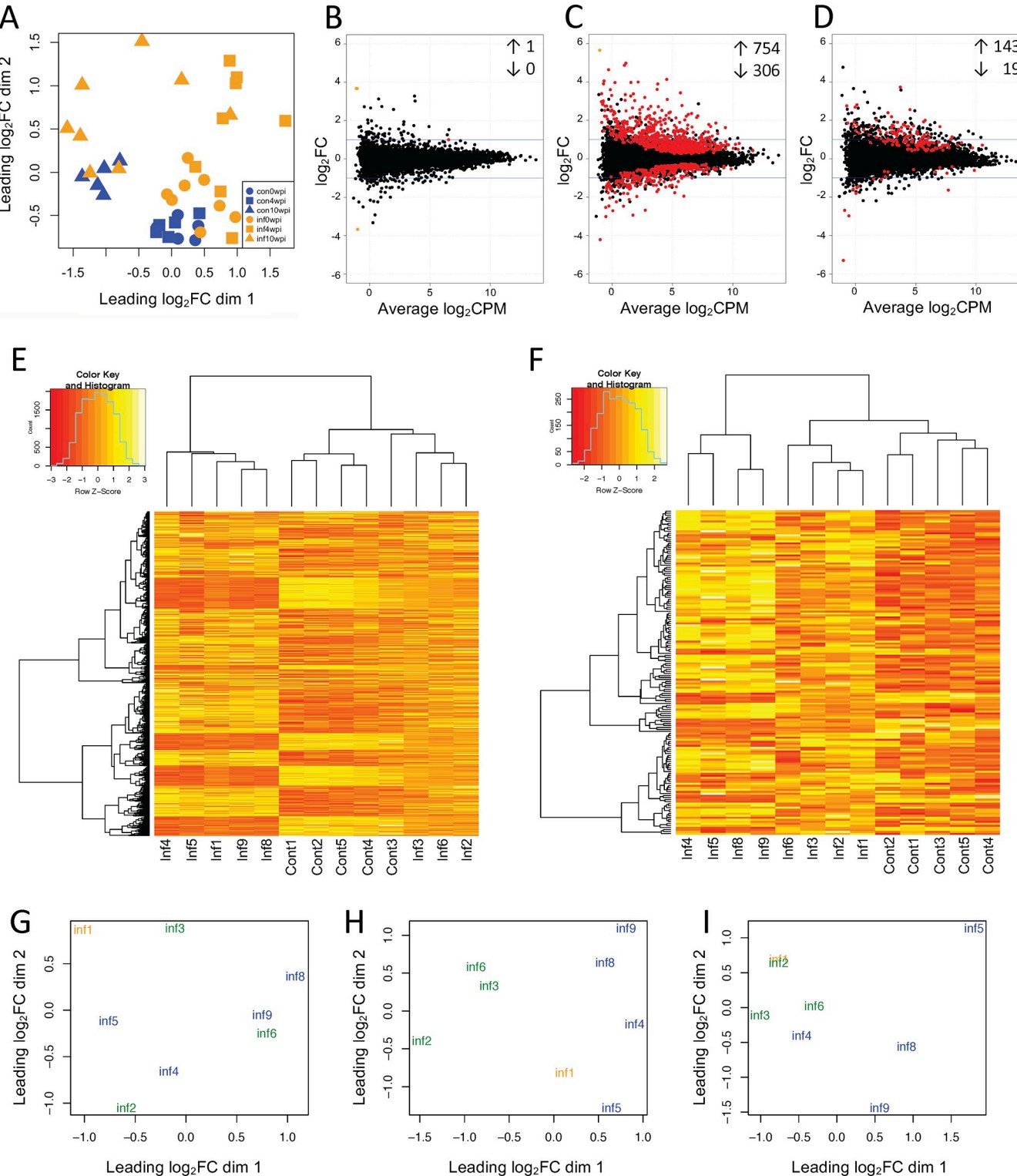

**Fig 2. Differential gene expression observed between *M. bovis*-infected and uninfected cattle revealed two transcriptional clusters of infected animals.** Whole blood from *M. bovis* experimentally infected (n = 8) and uninfected (n = 5) cattle at 0 wpi, 4 wpi, and 10 wpi was collected and analyzed for differential gene expression. (A) MDS plot of infected and uninfected samples at 0 wpi, 4 wpi, and 10 wpi based on the top 500 genes with highest standard deviations between treatment groups. Each point represents an individual sample, and smaller distances between points represent greater similarities. (B-D) Gene expression in infected compared to uninfected cattle at 0 wpi (B), 4 wpi (C), and 10 wpi (D). Positive logFC indicates genes with increased expression in infected samples, while negative logFC indicates genes with decreased expression in infected samples. Each point represents an individual gene. Red points

indicate differential gene expression reaching statistical significance (FDR < 0.05); orange points correspond to genes with zero counts in all samples of one treatment group. Black points indicate genes that were not differentially expressed and did not have all-zero counts in one treatment group. (E-F) Hierarchical clustering of samples based on 1,060 genes at 4 wpi (E) and 162 genes at 10 wpi (F) differentially expressed (FDR < 0.05) between infected and uninfected cattle. (G-I) MDS plots of infected animals only at 0 wpi (G), 4 wpi (H), and 10 wpi (I) based on the top 500 genes with highest standard deviations. Each point represents an individual sample, and smaller distances between points represent greater similarities. Samples from the 2 infected transcriptional clusters are denoted by green or blue text, while the sample clustering differently at each post-infection timepoint is denoted by orange text. n = 3 green (clustering more closely with uninfected samples), n = 4 blue (clustering more distantly from uninfected samples), and n = 1 sample that did not consistently cluster (Inf1).

Extrapulmonary lesions were also observed in the liver and/or kidneys in 3 of 8 calves (Fig 3C and Table 2). In all infected animals, viable *M. bovis* was recovered from all pulmonary lymph nodes, all lung lobes, and extrathoracic lesions. Although all calves had tuberculous lesions, based on lesion scores and pulmonary lymph node weights, cattle could be grouped into 2 categories: severely affected and moderately affected. Severely affected cattle had higher lymph node weights and lesion scores than did those classified as moderately affected (Fig 3D–3G and S1 Table). As with clinical presentation, differences in lesion severity correlated to the distinct transcriptional clustering observed. The severely affected group contained all 4 cattle that developed clinical signs and extrathoracic lesions. Collectively, these observations show a correlation between a transcriptional divergence from uninfected animals and disease severity.

## Transcriptional divergence following infection

Additional differential gene expression analysis was conducted at each timepoint to further assess differences between the two distinct clusters of infected cattle. Samples from animal Inf1 were excluded from all further cluster analysis because of the unmatched clustering observed for this animal at 4 wpi compared to 10 wpi. Both moderately- and severely-affected cattle were compared to uninfected cohorts to assess differential gene expression from a total of 14,525 genes (S2 File). When compared to uninfected controls, moderately-affected cattle showed only 35 (0.24%) differentially-expressed genes at 4 wpi, while no differential gene expression was observed at 0 wpi or 10 wpi (Fig 4A–4C and S2 File). Conversely, severely-affected cattle showed much greater differential gene expression, with a total of 8 (0.06%), 2,488 (17.13%) and 2,015 (13.87%) genes being differentially expressed at 0 wpi, 4 wpi, and 10 wpi, respectively, when compared to uninfected controls (Fig 4D–4F and S2 File). When compared to each other, moderately- and severely-affected cattle demonstrated 3 (0.02%), 604 (4.16%), and 916 (6.31%) genes differentially expressed genes at 0 wpi, 4 wpi, and 10 wpi, respectively (Fig 4G–4I and S2 File). Altogether, the data suggests that of the three groups (uninfected, moderately-, and severely-affected cattle), severely-affected cattle showed the most divergent transcriptional profile at 4 and 10 wpi. Additionally, direct comparison of moderately- and severely-affected cattle showed that a greater number of differentially-expressed genes was observed at 10 wpi compared to 4 wpi, suggesting that the 2 groups became increasingly divergent later in infection.

**Table 1. Clinical signs of cattle inoculated via aerosol with 1.77 X 10$^6$ CFU of *M. bovis* strain 10–7428.**

| Clinical Signs | Animal Number | | | | | | | |
|---|---|---|---|---|---|---|---|---|
| | Inf9 | Inf8 | Inf6 | Inf5 | Inf4 | Inf3 | Inf2 | Inf1 |
| Coughing | Yes | Yes | No | Yes | Yes | No | No | No |
| Labored breathing | Yes | Yes | No | No | No | No | No | No |
| Febrile | Yes | Yes | No | No | No | No | No | No |
| Time of Necropsy (PID) | 77 | 78 | 165 | 294 | 301 | 355 | 361 | 370 |

PID: post infection day.

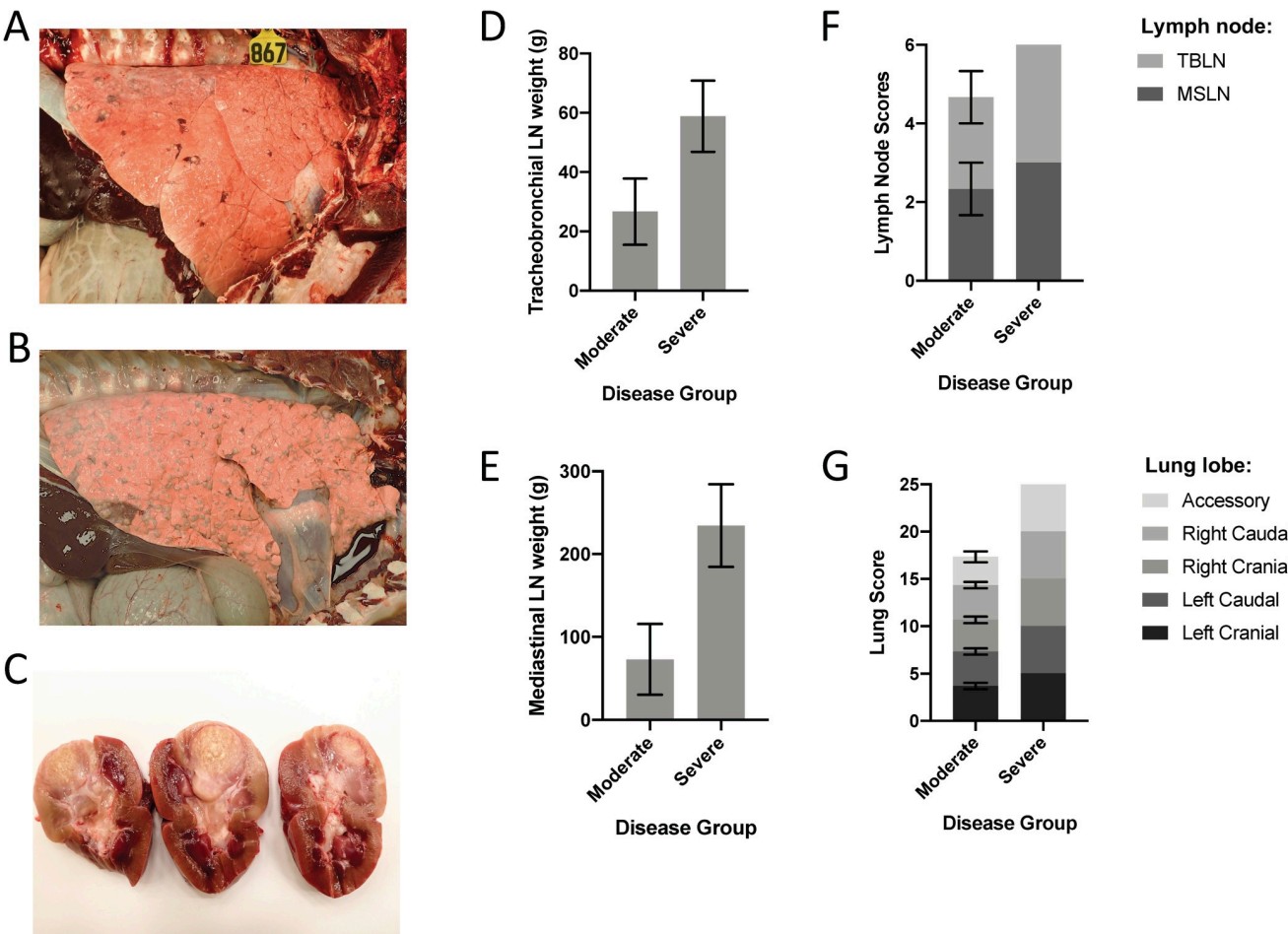

**Fig 3. Cattle experimentally infected with *M. bovis* develop moderate or severe disease.** (A-C) Gross pathology of *M. bovis*-infected cattle. Lung from moderately-affected (A) or severely-affected (B) calf following experimental infection. Kidney with tuberculous lesions from extrapulmonary dissemination in a severely-affected calf (C). (D-E) Weight of tracheobronchial (D) and mediastinal (E) lymph nodes from moderately-affected (n = 3) vs. severely-affected (n = 4) cattle. Black bars represent standard error of the mean. (F-G) Disease pathology scores of respiratory-associated lymph nodes (F) and lung lobes (G) in moderately- (n = 3) vs. severely-affected (n = 4) cattle. Black bars represent standard error of the mean. LN = lymph node; TBLN = tracheobronchial lymph node; MSLN = mediastinal lymph node.

Lists of differentially expressed genes between infected and uninfected cattle at 4 wpi and 10 wpi were compared to further characterize the transcriptional differences between moderately- and severely-affected cattle. At 4 wpi, 25 genes had commonly increased expression in both moderately- and severely-affected cattle when compared to uninfected cattle (Fig 5A and S3 File). When comparing severely-affected to uninfected cattle at both 4 and 10 wpi, we also observed common increased gene expression of 791 genes (Fig 5A and S3 File) and common decreased gene expression of 408 genes (Fig 5B and S4 File).

These comparisons indicate that while some genes are differentially expressed following infection with *M. bovis*, a more distinct transcriptional profile between uninfected and infected animals is observed in animals with clinical disease.

## Innate immune bias in severely-affected cattle

Enrichment analyses were performed using lists of genes with increased or decreased expression. This allowed us to summarize the biological implications of the transcriptional

**Table 2. Lesion scores and lymph node weights from cattle inoculated via aerosol with 1.77 X 10$^6$ CFU *M. bovis* strain 10–7428.**

| Group Assignment | Severe | Severe | Severe | Severe | Moderate | Moderate | Moderate | Undetermined |
|---|---|---|---|---|---|---|---|---|
| Tissue | Inf9 | Inf8 | Inf5 | Inf4 | Inf6 | Inf3 | Inf2 | Inf1 |
| Tracheobronchial LN | 3 | 3 | 3 | 3 | 1 | 3 | 3 | 3 |
| Mediastinal LN | 3 | 3 | 3 | 3 | 1 | 3 | 3 | 3 |
| Right Cranial Lobe | 5 | 5 | 5 | 5 | 3 | 4 | 3 | 4 |
| Right Caudal Lobe | 5 | 5 | 5 | 5 | 3 | 4 | 4 | 4 |
| Left Cranial Lobe | 5 | 5 | 5 | 5 | 3 | 4 | 4 | 4 |
| Left Caudal Lobe | 5 | 5 | 5 | 5 | 3 | 4 | 4 | 4 |
| Accessory Lobe | 5 | 5 | 5 | 5 | 2 | 4 | 3 | 4 |
| Total LN Score | 6 | 6 | 6 | 6 | 2 | 6 | 6 | 6 |
| Total Lung Score | 25 | 25 | 25 | 25 | 14 | 20 | 18 | 20 |
| Total Lesion Score | 31 | 31 | 31 | 31 | 16 | 26 | 24 | 26 |
| Mediastinal LN wt (g) | 360.5 | 170.8 | 266.1 | 140.4 | 18.2 | 157.2 | 43.5 | 123.0 |
| Tracheobronchial LN wt (g) | 90.65 | 55.0 | 57.2 | 32.4 | 11.4 | 48.4 | 20.2 | 58.1 |
| Extrapulmonary lesions | Yes | Yes | No | Yes | No | No | No | No |

LN = lymph node.

differences observed between uninfected, moderately- and severely-affected groups at 4 and 10 wpi. We were unable to analyze genes with increased expression at 10 wpi or decreased expression at 4 wpi or 10 wpi in moderately-affected compared to uninfected cattle due to low numbers or the absence of differentially expressed genes. Top biological processes enriched in severely-affected cattle compared to uninfected (S1 Fig and S5 File) and to moderately-affected (S2 Fig and S5 File) were processes related to host defense and immune response that were highly overlapping at 4 wpi and 10 wpi. These same processes were absent in moderately-affected cattle compared to uninfected (S1 Fig and S5 File) cattle at 4 wpi and compared to severely-affected cattle (S2 Fig and S5 File) at both 4 wpi and 10 wpi, as well as in uninfected compared to severely-affected cattle at both post-infection timepoints (S3 Fig and S5 File).

Since many of the biological processes enriched in severely-affected cattle were associated with host defense and the immune response, gene sets were reanalyzed for enrichment of only immune-related processes to further examine immune associations. Immune processes enriched in moderately-affected compared to uninfected animals at 4 wpi involved regulation of T cell proliferation. Immune processes enriched in severely- affected compared to uninfected animals involved innate immune pathways and responses to interferon-gamma (Fig 6A and S6 File). Only one immune process was enriched in uninfected compared to severely-affected cattle and involved type IV hypersensitivity at 10 wpi (S6 File). By comparing moderately- to severely-affected cattle at post-infection timepoints, enrichment of immune processes relating to T cells, neutrophils, complement, and cytokine production was observed in moderately-affected cattle, while responses to interferon-gamma were enriched in severely-affected cattle (Fig 6B and S6 File). Cell type enrichment further elucidated the immune shifts observed in moderately- and severely-affected cattle at post-infection timepoints. Severely-affected cattle showed enrichment of innate immune cells compared to either moderately-affected or uninfected cohorts, while T cells were enriched in uninfected compared to severely-affected cattle (Fig 6C and S2 Table). These results indicate severely-affected cattle have a transcriptional bias towards innate immune processes and innate cell types compared to moderately-affected or uninfected cohorts.

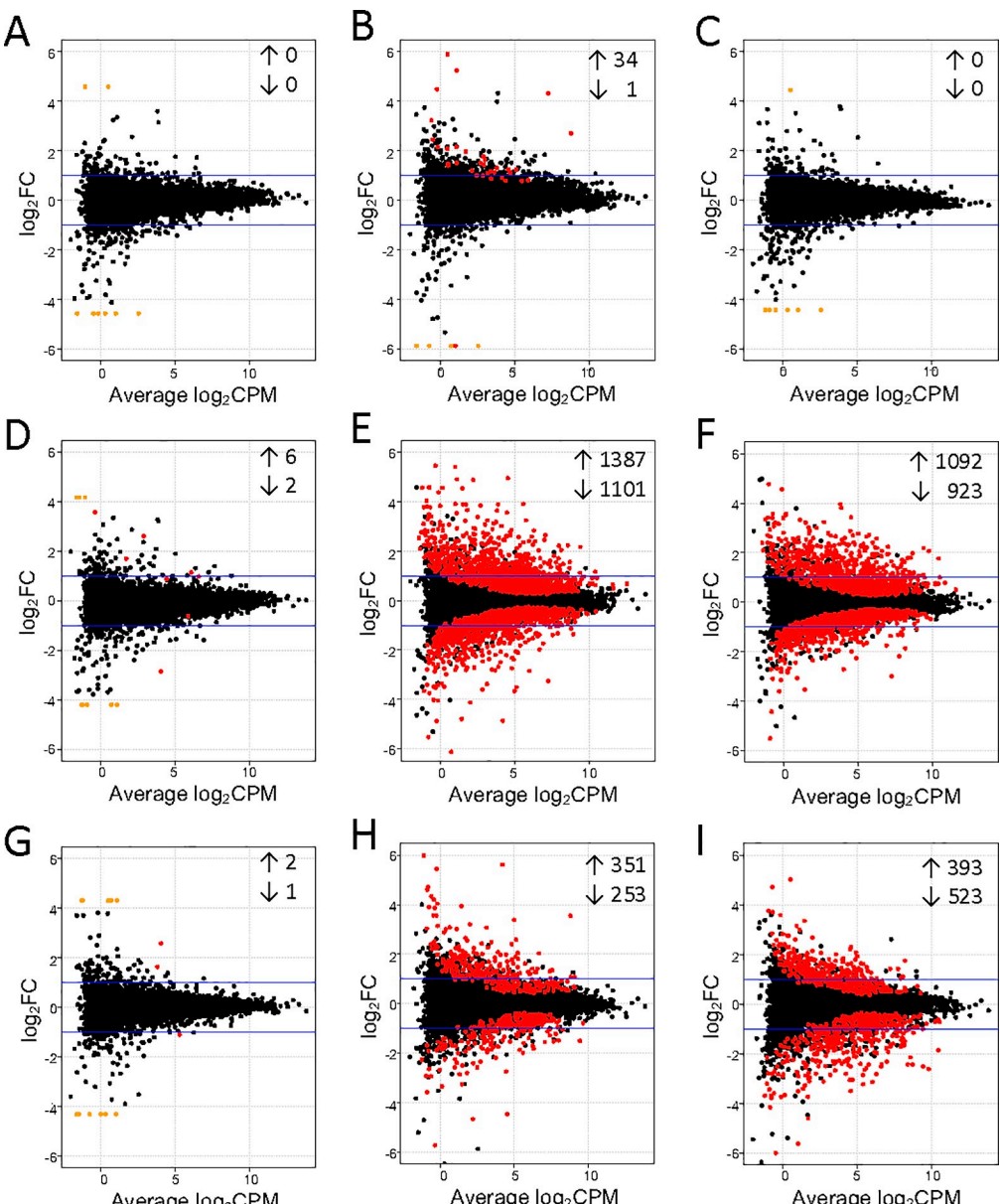

**Fig 4. Clusters of infected cattle showed differential gene expression compared to uninfected cohorts and diverge from each other over time.** Whole blood from severely-affected (n = 4), moderately-affected (n = 3), and uninfected (n = 5) cattle at 0 wpi, 4 wpi, and 10 wpi was collected and analyzed for differential gene expression. Gene expression in moderately-affected compared to uninfected cattle (A-C) at 0 wpi (A), 4 wpi (B), and 10 wpi (C); in severely-affected compared to uninfected cattle (D-F) at 0 wpi (D), 4 wpi (E), and 10 wpi (F); and in moderately-affected compared to severely-affected cattle (G-I) at 0 wpi (G), 4 wpi (H), and 10 wpi (I). Positive logFC indicates greater expression in infected samples (A-F) or moderately-affected samples (G-I), while negative logFC indicates greater expression in infected samples (A-F) or severely-affected samples (G-I). Each point represents an individual gene. Red points indicate significant differential gene expression (FDR < 0.05); orange points correspond to genes with zero counts in all samples of one treatment group. Black points indicate genes that were not differentially expressed and did not have all-zero counts in one treatment group.

## Discussion

This study characterizes the transcriptional changes that occur in peripheral blood leukocytes in early weeks following *M. bovis* infection in cattle. We found three distinct transcriptional

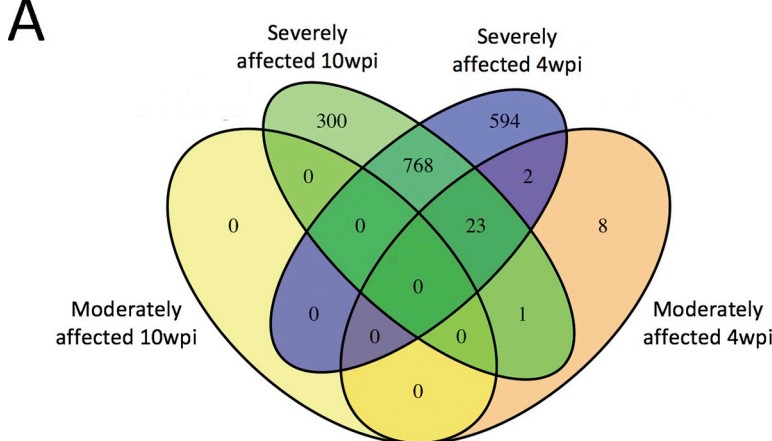

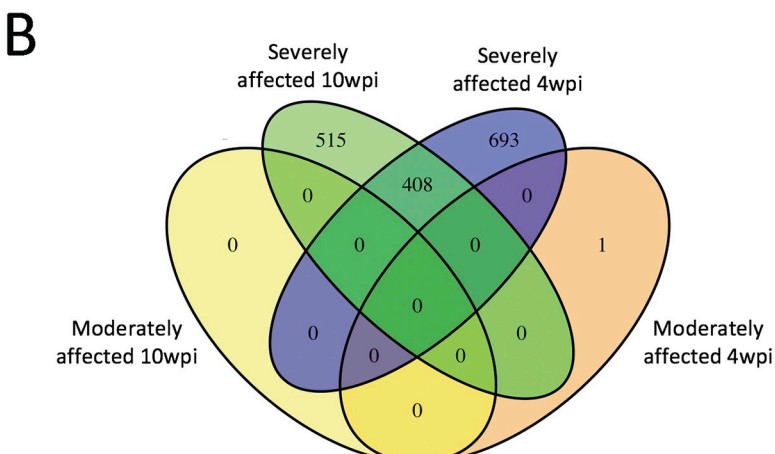

**Fig 5. Common differential gene expression in infected clusters at 4 wpi.** Venn diagram of differentially expressed genes with increased (A) and decreased (B) expression between moderately or severely affected cattle compared to uninfected cattle at 4 wpi and 10 wpi. Uninfected cattle n = 5, moderately affected cattle n = 3, severely affected cattle n = 4. wpi = weeks post-infection.

profiles: one in uninfected animals, and two within infected animals, which corresponded to clinical presentation of disease (i.e. moderately-affected and severely-affected animals).

Moderately-affected animals did not show any clinical signs of disease, such as coughing, labored breathing, or pyrexia. On post-mortem evaluation, we observed lesions consistent with *M. bovis* infection, which were found primarily in the lungs and in lung-associated lymph nodes. The transcriptional profile of these animals closely resembled that of uninfected cohorts, especially at the later 10 wpi timepoint. This would suggest that for this experimental model, early timepoints after infection may be a critical window for detecting the transcriptional biomarkers of infection. Information from very early timepoints following *M. bovis* infection could provide insight into transcriptional dynamics involved in the early interactions between mycobacteria and the host, and how this is associated with disease control versus progression.

In comparison, severely-affected animals displayed coughing, labored breathing and a febrile response as early as 35 days post-infection. Post-mortem examination revealed that these animals had increased lesion severity in the lungs and pulmonary lymph nodes, as well

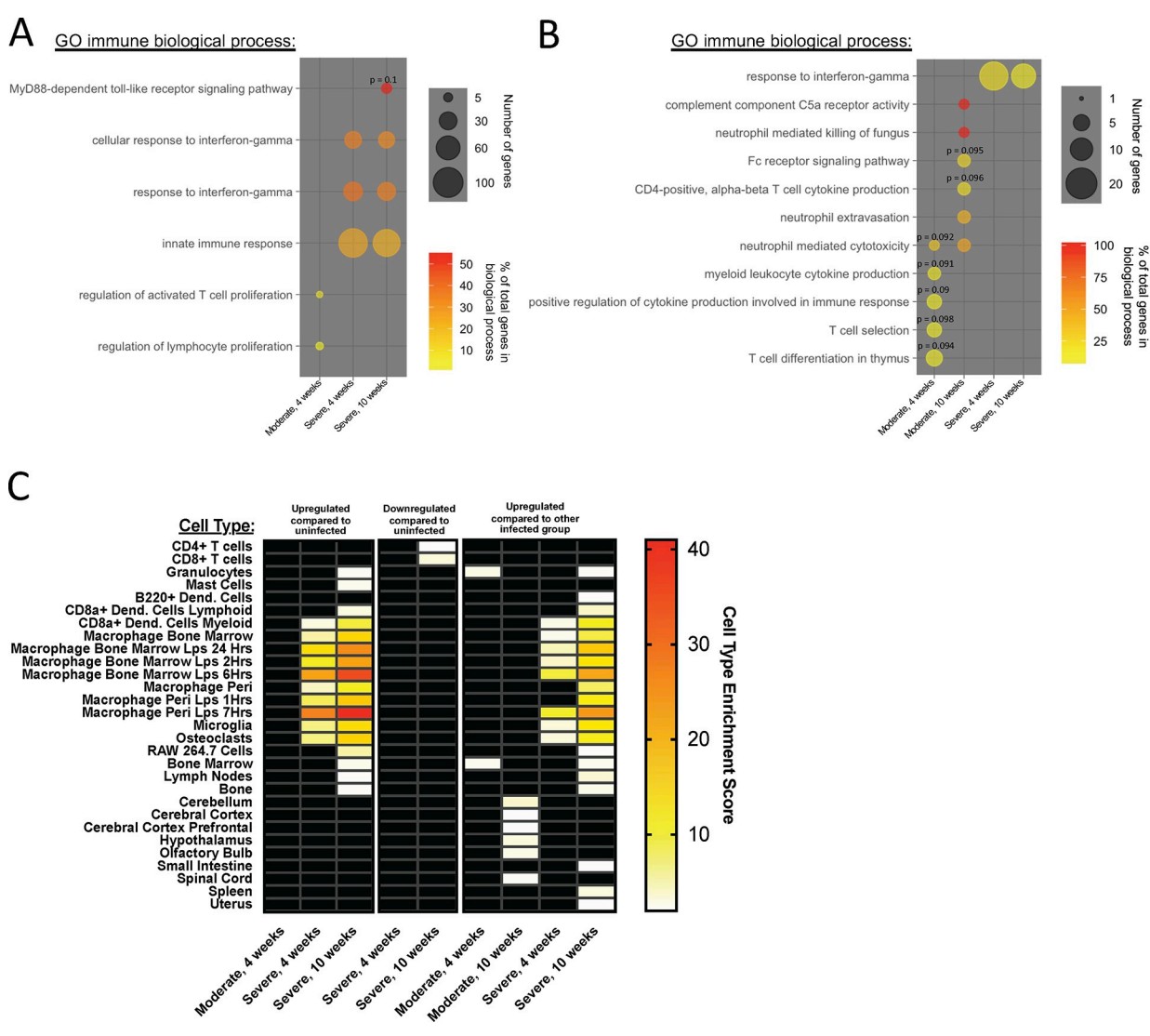

**Fig 6. Severely-affected cattle show biases towards innate immune biological processes and cell types.** (A-B) Significantly enriched immune biological processes in infected compared to uninfected cattle (A) and one cluster of infected cattle compared to the other infected cluster (B). Processes with corrected p-values < 0.1 were included. P-values are indicated for processes with a corrected p-value < 0.05. (C) Heat map of cell type enrichment scores from comparisons of uninfected, moderately-affected (moderate), and severely-affected (severe) cattle at 4 or 10 weeks post-infection. Cell types that showed significant enrichment in at least one of the comparisons are displayed on the y axis. Comparisons are denoted on the x axis. Heatmap color indicates cell enrichment scores, with larger values indicating higher enrichment scores. Cell enrichment scores greater than 2 were considered significant. Black boxes indicate cell enrichment scores less than 2. Uninfected cattle (n = 5), moderately-affected cattle (n = 3), severely-affected cattle (n = 4).

as extra-thoracic lesions in the liver and/or kidneys. The early onset of clinical signs observed in this study correlates with peripheral dissemination of *M. bovis*. Therefore, it is likely that the transcriptional profile observed in these animals, distinct from that of uninfected and moderately-affected animals, is related to extra-thoracic dissemination.

In this study, we investigated the *ex vivo* gene expression profile of leukocytes from whole blood samples from *M. bovis*-infected cattle using RNA-seq. Similar to previous findings from transcriptomic analysis of PBMC or PBL from infected and control cattle [10–12], we observed a clear transcriptional difference between uninfected and infected animals (regardless of clinical status) at 4 wpi, despite the lack of *in vitro* antigen stimulation. Enrichment analyses

indicated that the top biological processes enriched in infected cattle compared to uninfected cattle involved processes related to host defense and immune responses. This is consistent with previous work [11, 12]. However, in our data set, these differences diminished with time, and were minimal by 10 wpi. Furthermore, these transcriptional differences become less obvious if animals were segregated by clinical status, with moderately-affected animals becoming indiscriminate from controls by 10 wpi.

Severely-affected animals demonstrated the most transcriptionally distinct profile, as expected. In-depth analysis of pathways within host defense and immune response categories revealed an enrichment for upregulated genes associated with innate immune responses. This is in contrast to work that reported transcriptional suppression of genes involved in innate function following *M. bovis* infection in cattle [9, 10, 50]. Work in human and mouse models show that *M. bovis* down-modulates specific aspects of innate immunity including toll-like receptor (TLR)-mediated signaling [51–53], dendritic cell function [54] and antigen presentation by macrophages [55, 56]. However, despite this suppression in genes associated with innate immunity, infection with *M. bovis* also triggers a shift in the ratio of circulating monocytes to lymphocytes. This increase in lymphocyte numbers is usually correlated with protection [11]. Consistent with these findings, we observed that moderately-affected animals showed an enrichment of immune processes related to T cells, which was not seen in severely-affected animals; however, whether the enrichment can be attributed to increased proportions of circulating T cells, transcriptional alteration to existing cells, or a combination of both is unknown. It should be noted that, to our knowledge, the relationship between *M. bovis* infection and innate immune function has not been analyzed in the context of animals exhibiting clinical signs, as shown in this work. Our findings could suggest that, not unexpectedly, enhanced and sustained innate immune activation are detrimental to the host, and likely contribute to disease progression and bacterial dissemination. However, whether innate immune biases noted in our data are due to shifts in proportions of circulating cells, transcriptional alterations within the existing cells of circulation, or a combination of both is again unknown.

Further evidence of innate immune bias in severely-affected animals was observed in the cell type enrichment analysis. Severely-affected animals showed an enrichment in innate immune cells (primarily dendritic cells and macrophages), as compared to moderately-affected and uninfected cohorts. Interestingly, a study of transcriptional profiling of human patients with active tuberculosis showed there is a decrease in the abundance of B and T cell transcripts and an increase in myeloid-related transcripts, as compared to latently-infected patients [57]. Based on this report, there appears to be a correlation between clinical disease and innate immune bias. Lacking hematological information pertaining to cell subpopulations from our samples, we cannot determine if this enrichment in innate cells is due to transcriptional changes within cells or due to a change in cell numbers. Nevertheless, the observed transcriptional profile of clinical animals showing innate immune bias, and its similarities to human patients with active TB, provide insights into the immune responses that fail to control mycobacterial infection.

The infectious dose and aerosol route used in this study delivered a high dose of bacteria into the respiratory tract. Natural infection with *M. bovis* results in a slow-progressing disease, with animals showing limited clinical signs. In contrast, while all animals were inoculated at the same time and with the same inoculum preparation, clinical signs occurred only in some animals. At this time, we cannot explain why some animals developed more severe clinical disease as compared to others. While we cannot rule out the possibility of variability during the experimental infection process, this raises some important questions regarding individual susceptibility to infection. Furthermore, this variation allowed for an unique assessment of gene expression profiles in animals with distinct clinical presentations.

Experimental, high-dose infection may not fully recapitulate the transcriptional changes that occur following natural infection where the dose is likely lower and may involve multiple exposure events. However, the data obtained from this study indicates: 1) the quiescent nature of *M. bovis* infection, as moderately-affected animals appear transcriptionally similar to uninfected controls, 2) identifying biomarkers of infection presents a challenge, especially in subclinical animals, and 3) clinical severity has a unique transcriptional gene pathway profile, characterized by enhanced innate responses. To our knowledge, this is the first report of transcriptomic analysis of clinical cattle and the first to examine such early time-points.

## Supporting information

**S1 File.**
(XLSX)

**S2 File.**
(XLSX)

**S3 File.**
(XLSX)

**S4 File.**
(XLSX)

**S5 File.**
(XLSX)

**S6 File.**
(XLSX)

**S1 Table. Mean lesion scores and lymph node weights (g) with 95% confidence intervals.**
(DOCX)

**S2 Table. Cell type enrichment scores from comparisons of uninfected, moderately-affected, and severely-affected cattle.** Cell type enrichment scores were obtained from lists of differentially expressed genes. Scored > 2 were considered significant. AvB10down = enriched in severe compared to moderate at 10 wpi; AvB10up = enriched in moderate compared to severe at 10 wpi; AvB4down = enriched in severe compared to moderate at 4 wpi; AvB4up = enriched in moderate compared to severe at 4 wpi; BvC10down = enriched in control compared to severe at 10 wpi; BvC10up = enriched in severe compared to control at 10 wpi; BvC4down = enriched in control compared to severe at 4 wpi; BvC4up = enriched in severe compared to control at 4 wpi.
(DOCX)

**S1 Fig. Enriched biological processes in *M. bovis* infected compared to uninfected cattle.** Top 5 enriched biological processes from each group based on lowest p values. P-values all < 0.05. Uninfected cattle n = 5, moderately affected cattle n = 3, severely affected cattle n = 4.
(TIF)

**S2 Fig. Enriched biological processes in one cluster of *M. bovis* infected cattle compared to the other cluster of infected cattle.** Top 5 enriched biological processes from each group based on lowest p values. P-values all < 0.05. Uninfected cattle n = 5, moderately affected cattle n = 3, severely affected cattle n = 4.
(TIF)

**S3 Fig. Enriched biological processes in uninfected compared to *M. bovis* infected cattle.**
Top 5 enriched biological processes from each group based on lowest p values. P-values
all < 0.05. Uninfected cattle n = 5, moderately affected cattle n = 3, severely affected cattle
n = 4.
(TIF)

## Acknowledgments

We thank Shelly Zimmerman for unmatched laboratory assistance; Dr. Rebecca Cox and Dr.
Carly Kanipe for excellent veterinary care and assistance; Robin Zeisness, Dave Lubbers, Lisa
Ashburn, Dennis Johannes, and Kaitlyn Launderville for outstanding animal care and assis-
tance; Judi Stasko and Ginny Montgomery for superb histology services; and Dr. David Alt
and the Iowa State University DNA Facility for sample sequencing.

## Author Contributions

**Conceptualization:** Jayne E. Wiarda, W. Ray Waters, Tyler C. Thacker, Mitchell V. Palmer.

**Data curation:** Jayne E. Wiarda, Darrell O. Bayles.

**Formal analysis:** Jayne E. Wiarda.

**Investigation:** W. Ray Waters, Mitchell V. Palmer.

**Methodology:** Jayne E. Wiarda, W. Ray Waters, Tyler C. Thacker, Mitchell V. Palmer.

**Project administration:** W. Ray Waters, Tyler C. Thacker, Mitchell V. Palmer.

**Writing – original draft:** Jayne E. Wiarda, Paola M. Boggiatto, Mitchell V. Palmer.

**Writing – review & editing:** Jayne E. Wiarda, Paola M. Boggiatto, Darrell O. Bayles, Mitchell
V. Palmer.

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
