## [Decision Letter · Decision Letter 0]

19 May 2020

PONE-D-20-09070

Severity of bovine tuberculosis is associated with innate immune-biased transcriptional signatures of whole blood in early weeks after experimental Mycobacterium bovis infection

PLOS ONE

Dear Dr. Boggiatto,

Thank you for submitting your manuscript to PLOS ONE. After careful consideration, we feel that it has merit but does not fully meet PLOS ONE’s publication criteria as it currently stands. Therefore, we invite you to submit a revised version of the manuscript that addresses the points raised during the review process.

This is a well-designed, well-implemented and interesting study with the potential to provide new insights on early events following M.bovis infection of cattle. However, Reviewer 2 has highlighted several aspects which need addressing. In particular the FASTQ files generated for the RNA-seq analysis and Supplementary Files S1-S6 need to be supplied and it needs to be clarified which background gene set was used for GO term enrichment analyses. Other comments regarding interpretation of the data should also be corrected such as use of the terms “upregulated”, “downregulated” or “activated” when there is no direct evidence for these processes other than changes in gene expression. Please carefully respond to the other specific points made by reviewer 2.

We would appreciate receiving your revised manuscript by Jul 03 2020 11:59PM. To enhance the reproducibility of your results, we recommend that if applicable you deposit your laboratory protocols in protocols.io, where a protocol can be assigned its own identifier (DOI) such that it can be cited independently in the future. For instructions see: http://journals.plos.org/plosone/s/submission-guidelines#loc-laboratory-protocols

We look forward to receiving your revised manuscript.

Kind regards,

Ann Rawkins, PhD

Academic Editor

PLOS ONE

2. In your Methods section, please provide additional details regarding the animals used in your study and ensure you have described the source. For more information regarding PLOS' policy on materials sharing and reporting, see https://journals.plos.org/plosone/s/materials-and-software-sharing#loc-sharing-materials.

3. We note that you are reporting an analysis of a microarray, next-generation sequencing, or deep sequencing data set. PLOS requires that authors comply with field-specific standards for preparation, recording, and deposition of data in repositories appropriate to their field. Please upload these data to a stable, public repository (such as ArrayExpress, Gene Expression Omnibus (GEO), DNA Data Bank of Japan (DDBJ), NCBI GenBank, NCBI Sequence Read Archive, or EMBL Nucleotide Sequence Database (ENA)). In your revised cover letter, please provide the relevant accession numbers that may be used to access these data. For a full list of recommended repositories, see http://journals.plos.org/plosone/s/data-availability#loc-omics or http://journals.plos.org/plosone/s/data-availability#loc-sequencing.

Reviewers' comments:

Reviewer's Responses to Questions

**Comments to the Author**

1. Is the manuscript technically sound, and do the data support the conclusions?

Reviewer #1: Yes

Reviewer #2: Yes

2. Has the statistical analysis been performed appropriately and rigorously? 

Reviewer #1: Yes

Reviewer #2: Yes

3. Have the authors made all data underlying the findings in their manuscript fully available?

Reviewer #1: Yes

Reviewer #2: No

4. Is the manuscript presented in an intelligible fashion and written in standard English?

Reviewer #1: Yes

Reviewer #2: Yes

5. Review Comments to the Author

Reviewer #1: The study is a well thought out thorough investigation into the transcriptional response of cattle experimentally infected with M. bovis. It builds on other previous studies in this area but brings new insights regarding early time points and the correlation between transcriptional changes and the severity of infection.

Reviewer #2: Severity of bovine tuberculosis is associated with innate immune-biased transcriptional signatures of whole blood in early weeks after experimental Mycobacterium bovis infection

++++++++++++++++++++++++++++++++++++++++++++++++++++++++++++++++++++++++++++++++++++++++++++++++++++++++++++++

This is a well-designed, well-implemented and scientifically interesting study of transcriptional responses in Holstein cattle experimentally infected with Mycobacterium bovis.

I have several general points and then some specific points about the study and the manuscript describing the work.

General Points

+++++++++

The manuscript is well-written, easy to understand and scientifically authoritative. There are a small number of typographical errors that I detail in the Specific Points section.

This is a suggestion, but I think the overall experimental design, purpose and outputs from the study could be usefully represented in a summary overview diagram (new Figure 1) that shows the experimental workflow from start to finish. This could include the infection time course, the tissue sampling for post-mortem pathology, the RNA-seq analysis of peripheral blood leukocytes and the basic analyses of DE genes plus the downstream data mining using GO term categories and cell type enrichment etc. This would make it easier for the reader to understand exactly what was done and appreciate the importance of the work.

The FASTQ files generated for the RNA-seq analysis are not currently available from the NCBI SRA repository. There is a BioProject accession entry (PRJNA600004) but not data is available. This must be made available when the paper is published. It should really be made available to the manuscript Reviewers and this is possible through the NCBI BioProject/SRA system - see the following link:

https://www.ncbi.nlm.nih.gov/sra/docs/submitquestions/#question3gen

Supplementary Files S1-S6 are not available to Reviewers with the submitted manuscript files on the PLOS ONE Editorial Manager website. Most of these files presumably contain the "meat" of the RNA-seq results (e.g. gene ID, log2FC, P value, FDR-adjusted P value etc.), which need to be made available to Reviewers to properly assess the results obtained and the biological relevance of the differentially expressed genes and gene set enrichment analyses etc. Can the authors ensure that these files are available with revised version of the manuscript?

t is not really appropriate to use the terms “upregulated” or “downregulated” in the context of the results reported here. The peripheral blood leukocytes examined are a heterogeneous cell mixture and the authors do not have clear evidence that genes increased or decreased in expression are actively upregulated or downregulated (e.g. through chromatin state changes, transcription factors, microRNAs etc.). The changes could be due to changes in cell composition; therefore, it is better to use the terms “increased in expression” or “decreased in expression”. As it happens, the CTen paper (Shoemaker et al. 2012) has a good overview of why this is important.

Shoemaker J.E., et al. (2012) CTen: a web-based platform for identifying enriched cell types from heterogeneous microarray data. BMC Genomics 13, 460.

Specific Points

+++++++++

Line 54: The $3 billion dollar figure used for the global financial loss associated with bovine tuberculosis is 25 years old now (quarter of a century!). We’re all guilty of casually citing this reference, but at this stage it should really be qualified as a “conservative” or “long-standing” estimate.

Lines 102-103: Is there a specific Animal Ethics Committee approval code or number for this project?

Lines 223-224: I am surprised that the authors did not use the new bovine genome assembly for their reference genome (ARS-UCD1.2 - www.ncbi.nlm.nih.gov/assembly/GCF_002263795.1). This resource has been available for more than two years (since April 2018) and is now formally published in Gigascience. In our experience, it provides a much better reference genome than UMD3.1 for RNA-seq studies in cattle.

A more pedantic or bolshy Reviewer might insist that the analyses be re-done using the newer assembly. However, I would be interested to know the authors’ reasons/justification for not using ARS-UCD1.2.

Rosen B.D., et al. (2020) De novo assembly of the cattle reference genome with single-molecule sequencing. Gigascience 9.

Lines 253-266: The authors do not make it clear in this Methods section, but what background gene set did they used for these gene set GO term enrichment analyses? It is important that the appropriate background gene set is used, which should be the detectable expressed gene set, not the complete bovine transcriptome. Based on the results obtained in this study, the background set should be the 14,279 genes reported on line 304. Can the authors explicitly state what background gene set was used? If it is not the detectable gene set, they should consider re-doing the analyses because the GO enrichment results obtained using the complete bovine gene set as the background will be biased towards processes in peripheral blood leukocytes that may not be a consequence of M. bovis infection. See Timmons et al. 2015 for a more detailed explanation of why this is important.

Timmons J.A., et al. (2015) Multiple sources of bias confound functional enrichment analysis of global -omics data. Genome Biol. 16, 186.

Lines 263-266: Could the authors explain why the parameters for detection of enriched GO immune system processes were relaxed (e.g. P < 0.1 – is this an FDR-adjusted P value? – it is not clear). Also, there is a typo on line 265: “1” should be “a”.

Lines 308-331: Figure 1 and legend to Figure 1. The axes labels and text for Fig1B, Fig1C and Fig1D are far too small – they need to be increased in size to make them legible. The term “LogFC” is not specific enough and needs to be replaced with Log2FC (with appropriate subscripting) of “2” (in the legend and on the axes labels).

Lines 333-341 and elsewhere in the manuscript. It’s probably a long shot, but could a genetic explanation at least partially account for the two clusters of animals that exhibit distinct clinical and transcriptional profiles? For example, do the calves coded inf4, inf5, inf8 and inf9 share the same sire, with a genetic background that might account for low resilience to bovine TB and a concomitant severe disease clinical phenotype and expression profile? Although the MDS plot in Figure 1G does not indicate sharing of basal transcriptional profiles among inf4, inf5, inf8 and inf9 that could be due to close genetic relationship.

In this regard, it would be useful if the authors provided some more information in the Materials and Methods concerning the genetic relationships among the calves used for the study.

Lines 414-428: Figure 3 and legend to Figure 3. The axes labels and text for Fig3A to Fig3I are far too small – they need to be increased in size to make them legible. The term “LogFC” is not specific enough and needs to be replaced with Log2FC (with appropriate subscripting) of “2” (in the legend and on the axes labels).

Line 436: Typo: “File 3S” should be “File S3”.

Line 525: Reference 1 seems to be out of place here?

Line 535-538 and earlier lines: The following statement may not be correct.

“In-depth analysis of pathways within host defense and immune response categories revealed an enrichment for genes associated with innate immune responses. This is in contrast to work that reported transcriptional suppression of genes involved in innate function following M. bovis infection in cattle (9, 10, 51).”

Statistically significant enrichment of genes corresponding to particular biological processes does not necessarily correspond to activation (opposite of suppression) of innate immune responses. This may be because the enrichment is due to overrepresentation of genes that are decreased in expression in infected animals. Also, all of these analyses are suspect if the incorrect background gene set was used (see my previous point relating to the Timmons et al. 2015 Genome Biology paper).

The enrichment of genes associated with innate immune responses would indicate the opposite pattern to what was observed previously if the input data sets corresponded to genes exhibiting increased expression (note: not upregulation). The authors do not make it clear which input gene sets gave these results; was it just the set of genes exhibiting increased expression or was it the combined sets of genes exhibiting both increased and decreased expression? The authors need to be clearer on how they describe these different input data sets. It’s not even clear whether they segregated genes into two lists (increased in expression and decreased in expression), or whether it was a single list containing genes showing both increased and decreased expression in infected animals versus control non-infected animals (lines 450-451). From lines 254-256, it seems that it was the just the combined list of DE genes (FDR-adjusted P value <0.05).

Line 545: Again, enrichment of immune processes related to T cells does not necessarily mean that these processes are “activated”. For example, the enrichment could be due to genes that are both increased and decreased in expression in infected animals compared to control non-infected animals.

Line 550: “innate immune activation” – again, unless the input gene sets were only those exhibiting increased expression, then this statement cannot be supported by the data.

Overall, the point I am trying to make here is that “Enrichment” of DE genes in particular GO term categories or biological pathways does not automatically mean “Activation”.

It would be much easier for the Reviewer to evaluate these results and look at specific sets of genes if the Supplementary Files S1-S6 were actually available on the PLOS ONE Editorial Manager website. This absence of supporting files (and the possibility that the gene set GO term enrichment analyses were not performed with the appropriate background gene set) is the reason I selected "Major Revision" as my recommendation.

I have indicated that the statistical analyses have been conducted appropriately in the single pull-down menu because the statistical analyses of DE genes using RNA-seq data and the post-mortem pathology data have been performed correctly. It's only the gene set GO term enrichment that might be suspect.

6. PLOS authors have the option to publish the peer review history of their article (what does this mean?). If published, this will include your full peer review and any attached files.

Reviewer #1: Yes: Sharon Louise Kendall

Reviewer #2: Yes: David E. MacHugh

---

## [Author Response · Author response to Decision Letter 0]

25 Jun 2020

Dear PLOS ONE Editor, 

We thank you for your review of the manuscript entitled “Severity of bovine tuberculosis is associated with innate immune-biased transcriptional signatures of whole blood in early weeks after experimental Mycobacterium bovis infection.” We have addressed all changes requested either in the manuscript and/or through further clarification in this letter. 

Journal requirements

1. Please ensure that your manuscript meets PLOS ONE’s style requirements, including those for file naming.

We have edited the manuscript to make sure that all requirements were met. 

2. In your methods section, please provide additional details regarding the animals used in your study and ensure you have described the source.

We have added further information on the source of the animals used in the study. 

3. We note that you are reporting an analysis of a microarray, next-generation sequencing, or deep sequencing data sets. PLOS requires that authors comply with field-specific standards for preparation, recording and deposition of data in repositories appropriate to their field. Please upload these data to a stable, public repository…please provide the relevant accession numbers that may be used to access these data.

The data sets have been uploaded to a repository and the accession number is provided. Unfortunately, a delay had been placed on the data availability, but that issues has now been resolved. This information included in the original manuscript, under Materials and Methods, Data Availability is the correct information and the data should now be accessible. 

Reviewer Comments

"This is a suggestion, but I think the overall experimental design, purpose and outputs from the study could be usefully represented in a summary overview diagram (new Figure 1) that shows the experimental workflow from start to finish. This could include the infection time course, the tissue sampling for post-mortem pathology, the RNA-seq analysis of peripheral blood leukocytes and the basic analyses of DE genes plus the downstream data mining using GO term categories and cell type enrichment etc. This would make it easier for the reader to understand exactly what was done and appreciate the importance of the work."

Thank you for this suggestion. We have created a new Figure 1 depicting our experimental outline. All other figure numbers have also been changed throughout the text, accordingly.

"The FASTQ files generated for the RNA-seq analysis are not currently available from the NCBI SRA repository. There is a BioProject accession entry (PRJNA600004) but not data is available. This must be made available when the paper is published. It should really be made available to the manuscript Reviewers and this is possible through the NCBI BioProject/SRA system - see the following link:https://www.ncbi.nlm.nih.gov/sra/docs/submitquestions/#question3gen."

We thank the reviewer for their comment. The data files were uploaded and the accession number is correct. Unfortunately, we had a hold on the file accessibility, and therefore, we apologize for not having the data available at the time of the review. 

"Supplementary Files S1-S6 are not available to Reviewers with the submitted manuscript files on the PLOS ONE Editorial Manager website. Most of these files presumably contain the "meat" of the RNA-seq results (e.g. gene ID, log2FC, P value, FDR-adjusted P value etc.), which need to be made available to Reviewers to properly assess the results obtained and the biological relevance of the differentially expressed genes and gene set enrichment analyses etc. Can the authors ensure that these files are available with revised version of the manuscript?"

We apologize these files not being available during the initial submission of this manuscript. This was an unfortunate mistake, and we apologize for that. The figures are now available. 

"It is not really appropriate to use the terms “upregulated” or “downregulated” in the context of the results reported here. The peripheral blood leukocytes examined are a heterogeneous cell mixture and the authors do not have clear evidence that genes increased or decreased in expression are actively upregulated or downregulated (e.g. through chromatin state changes, transcription factors, microRNAs etc.). The changes could be due to changes in cell composition; therefore, it is better to use the terms “increased in expression” or “decreased in expression”. As it happens, the CTen paper (Shoemaker et al. 2012) has a good overview of why this is important."

We thank the reviewer for their comment, as it is a critical distinction to make. We have taken the advice and made the necessary adjustments throughout the text and we hope these changes are suitable. 

"Line 54: The $3 billion dollar figure used for the global financial loss associated with bovine tuberculosis is 25 years old now (quarter of a century!). We’re all guilty of casually citing this reference, but at this stage it should really be qualified as a “conservative” or “long-standing” estimate."

We thank the reviewer for this comment, however, we have not been able to find a better estimate of the global economic burden for bovine tuberculosis. We agree that this is a rather old estimate now, so we have made a modification in the text to reflect this in the manuscript. 

"Lines 102-103: Is there a specific Animal Ethics Committee approval code or number for this project?"

We have now provided the protocol number associated with this research study.

"Lines 223-224: I am surprised that the authors did not use the new bovine genome assembly for their reference genome (ARS-UCD1.2 - www.ncbi.nlm.nih.gov/assembly/GCF_002263795.1). This resource has been available for more than two years (since April 2018) and is now formally published in Gigascience. In our experience, it provides a much better reference genome than UMD3.1 for RNA-seq studies in cattle. A more pedantic or bolshy Reviewer might insist that the analyses be re-done using the newer assembly. However, I would be interested to know the authors’ reasons/justification for not using ARS-UCD1.2.Rosen B.D., et al. (2020) De novo assembly of the cattle reference genome with single-molecule sequencing. Gigascience 9."

We appreciate the reviewer’s recognition of this fact. The reason for not having used the new bovine genome assembly is that the initial processing of RNA-seq reads to obtain gene counts and do general infected versus uninfected DGE analysis was performed in early 2018, before the new genome was made available. It was not only until later that we received all pathology and culture results to continue analysis and writing.

"Lines 253-266: The authors do not make it clear in this Methods section, but what background gene set did they used for these gene set GO term enrichment analyses? It is important that the appropriate background gene set is used, which should be the detectable expressed gene set, not the complete bovine transcriptome. Based on the results obtained in this study, the background set should be the 14,279 genes reported on line 304. Can the authors explicitly state what background gene set was used? If it is not the detectable gene set, they should consider re-doing the analyses because the GO enrichment results obtained using the complete bovine gene set as the background will be biased towards processes in peripheral blood leukocytes that may not be a consequence of M. bovis infection. See Timmons et al. 2015 for a more detailed explanation of why this is important."

Thank you for your insight. We do state in lines 265-266 of the methods that the reference list of genes used was only genes that survived filtering during DGE analysis. This then would be our list of 14,525 genes found in File S2 that were used for DGE between severely affected, moderately affected, and uninfected animals. This list of genes is mentioned in line 458 as the filtered genes used for this DGE analysis. Please let us know if we can make this any clearer to the reader.

"Timmons J.A., et al. (2015) Multiple sources of bias confound functional enrichment analysis of global -omics data. Genome Biol. 16, 186."

The gene set used for the GO term enrichment analyses was indeed the filtered genes, not the complete bovine transcriptome. This has been revised in the Materials and Methods for clarification. 

"Lines 263-266: Could the authors explain why the parameters for detection of enriched GO immune system processes were relaxed (e.g. P < 0.1 – is this an FDR-adjusted P value? – it is not clear). Also, there is a typo on line 265: “1” should be “a”."

We thank the reviewer for their comment. Yes, it is an FDR corrected P value. We relaxed the parameters for detection since we did not have a lot of significant FDRs. However, we felt these findings could still be biologically meaningful, even if not statistically significant. It should be noted that we do denote these with exact p-values as a cautionary measure for the reader.

"Lines 308-331: Figure 1 and legend to Figure 1. The axes labels and text for Fig1B, Fig1C and Fig1D are far too small – they need to be increased in size to make them legible. The term “LogFC” is not specific enough and needs to be replaced with Log2FC (with appropriate subscripting) of “2” (in the legend and on the axes labels)."

We thank the reviewer for this feedback. The figure has been adjusted in order to make the legible. 

"Lines 333-341 and elsewhere in the manuscript. It’s probably a long shot, but could a genetic explanation at least partially account for the two clusters of animals that exhibit distinct clinical and transcriptional profiles? For example, do the calves coded inf4, inf5, inf8 and inf9 share the same sire, with a genetic background that might account for low resilience to bovine TB and a concomitant severe disease clinical phenotype and expression profile? Although the MDS plot in Figure 1G does not indicate sharing of basal transcriptional profiles among inf4, inf5, inf8 and inf9 that could be due to close genetic relationship. In this regard, it would be useful if the authors provided some more information in the Materials and Methods concerning the genetic relationships among the calves used for the study."

We thank the reviewer for this observation, as we too had some thoughts about the individual genetic susceptibility to infection. We make a brief mention of this in the discussion, lines 667-668. Unfortunately, we do not have individual calf information in terms of genetic relationships, as we do not breed animals on site. Since this study was performed several years ago, we are unsure whether or not we could obtain this information. We agree with the reviewer that genetic relationships may provide an interesting perspective on the data. However, even if we had this information, we feel we do not have a large enough number of calves in the analysis in order to make such conclusions. 

"Lines 414-428: Figure 3 and legend to Figure 3. The axes labels and text for Fig3A to Fig3I are far too small – they need to be increased in size to make them legible. The term “LogFC” is not specific enough and needs to be replaced with Log2FC (with appropriate subscripting) of “2” (in the legend and on the axes labels)."

As above, thank you for the comment and we have adjusted the labels accordingly.

"Line 436: Typo: “File 3S” should be “File S3”."

Typo has been changed in the text. 

"Line 525: Reference 1 seems to be out of place here?"

We apologize for the oversight. This has been fixed. 

"Line 535-538 and earlier lines: The following statement may not be correct. “In-depth analysis of pathways within host defense and immune response categories revealed an enrichment for genes associated with innate immune responses. This is in contrast to work that reported transcriptional suppression of genes involved in innate function following M. bovis infection in cattle (9, 10, 51).”

Statistically significant enrichment of genes corresponding to particular biological processes does not necessarily correspond to activation (opposite of suppression) of innate immune responses. This may be because the enrichment is due to overrepresentation of genes that are decreased in expression in infected animals. Also, all of these analyses are suspect if the incorrect background gene set was used (see my previous point relating to the Timmons et al. 2015 Genome Biology paper). The enrichment of genes associated with innate immune responses would indicate the opposite pattern to what was observed previously if the input data sets corresponded to genes exhibiting increased expression (note: not upregulation). The authors do not make it clear which input gene sets gave these results; was it just the set of genes exhibiting increased expression or was it the combined sets of genes exhibiting both increased and decreased expression? The authors need to be clearer on how they describe these different input data sets. It’s not even clear whether they segregated genes into two lists (increased in expression and decreased in expression), or whether it was a single list containing genes showing both increased and decreased expression in infected animals versus control non-infected animals (lines 450-451). From lines 254-256, it seems that it was the just the combined list of DE genes (FDR-adjusted P value <0.05)."

We thank the reviewer for their comment. We have tried to clarify that we used either lists of genes with increased expression or lists of genes with decreased expression for inputs of GO analysis. We have added text to clarify this in lines 258-259 of the Methods. Additionally, we made this clear in the discussion, line 626, and 637-639, to better reflect our findings. These individual gene lists are also available in File S5 and specify increased or decreased expression between the various pairwise comparisons. 

"Line 545: Again, enrichment of immune processes related to T cells does not necessarily mean that these processes are “activated”. For example, the enrichment could be due to genes that are both increased and decreased in expression in infected animals compared to control non-infected animals."

We have considered this comment and carefully reworded our conclusions to avoid making assumptions from our data. Please refer to lines 632-636 of the revised manuscript.

"Line 550: “innate immune activation” – again, unless the input gene sets were only those exhibiting increased expression, then this statement cannot be supported by the data."

In line with the previous comment, we have adjusted our wording to avoid making assumptions. Please see lines 639-644.

"Overall, the point I am trying to make here is that “Enrichment” of DE genes in particular GO term categories or biological pathways does not automatically mean “Activation”."

Thank you for this observation. We have tried to adjust our wording and conclusions to be aware of this in our revised manuscript.

"It would be much easier for the Reviewer to evaluate these results and look at specific sets of genes if the Supplementary Files S1-S6 were actually available on the PLOS ONE Editorial Manager website. This absence of supporting files (and the possibility that the gene set GO term enrichment analyses were not performed with the appropriate background gene set) is the reason I selected "Major Revision" as my recommendation."

We apologize for the lack of availability of these files. All supplementary material has been made available. We are unsure why all supplementary information was not available, but these files will be submitted with the resubmission. 

"I have indicated that the statistical analyses have been conducted appropriately in the single pull-down menu because the statistical analyses of DE genes using RNA-seq data and the post-mortem pathology data have been performed correctly. It's only the gene set GO term enrichment that might be suspect."

Thank you for this. We have addressed the reviewer’s concerns about our GO term enrichment in previous comments.

Again, we thank the editor and reviewers for their time and effort in reviewing our manuscript. We appreciate the comments and suggestions, which can only strengthen and make the message we are trying to convey clearer. We hope that we have addressed all concerns either in the text and/or through our explanations in this letter and hope that you will find this manuscript suitable for publication. 

Thank you, 

Paola M. Boggiatto

---

## [Decision Letter · Decision Letter 1]

30 Jul 2020

PONE-D-20-09070R1

Severity of bovine tuberculosis is associated with innate immune-biased transcriptional signatures of whole blood in early weeks after experimental Mycobacterium bovis infection

PLOS ONE

Dear Dr. Boggiatto,

Thank you for submitting your manuscript to PLOS ONE. After careful consideration, we feel that it has merit but does not fully meet PLOS ONE’s publication criteria as it currently stands. Therefore, we invite you to submit a revised version of the manuscript that addresses the points raised during the review process.

Please address the minor comments raised by Reviewer 2

We look forward to receiving your revised manuscript.

Kind regards,

Ann Rawkins, PhD

Academic Editor

PLOS ONE

Reviewers' comments:

Reviewer's Responses to Questions

**Comments to the Author**

1. If the authors have adequately addressed your comments raised in a previous round of review and you feel that this manuscript is now acceptable for publication, you may indicate that here to bypass the “Comments to the Author” section, enter your conflict of interest statement in the “Confidential to Editor” section, and submit your "Accept" recommendation.

Reviewer #1: All comments have been addressed

Reviewer #2: (No Response)

2. Is the manuscript technically sound, and do the data support the conclusions?

Reviewer #1: Yes

Reviewer #2: Yes

3. Has the statistical analysis been performed appropriately and rigorously? 

Reviewer #1: Yes

Reviewer #2: Yes

4. Have the authors made all data underlying the findings in their manuscript fully available?

Reviewer #1: Yes

Reviewer #2: Yes

5. Is the manuscript presented in an intelligible fashion and written in standard English?

Reviewer #1: Yes

Reviewer #2: Yes

6. Review Comments to the Author

Reviewer #1: I had no major comments on the manuscript but reviewer 2 highlighted deficiencies that needed to be addressed. These have been addressed.

Reviewer #2: The authors have improved the manuscript and associated files significantly in Revision 1.

The new Figure 1 is an excellent graphical representation of the work and make it much easier for readers to understand how the study was conducted.

There are just several small issues the authors should correct in a minor revision.

Line 54-55: This statement doesn't currently make any sense. The citation is from 1995 and the year being referred to 2004. Is there a citation for the $50 million cattle statistic? If so, the text could be re-worded as follows - note insertion of new citation and "at least" before "$3 billion"

"In 2004, it was estimated that worldwide, an estimated >50 million cattle are infected with M. bovis (CITATION), resulting in a financial loss of at least $3 billion USD annually (6)."

Lines 258 and 259: Please make it clear you are using Log2FC values (with subscript "2"). "LogFC" is not sufficient - a non-expert reader will assume this is Log10.

Lines 275 and 276: Same as previous comment.

Supplementary Files 3 and 4: The terms "upregulated" and "downregulated" are still being used inappropriately in these files. Please change to "increased expression" or "decreased expression" etc. as detailed in the review of the first version of the manuscript.

PLEASE NOTE: I do not need to check that these minor edits are completed. The PLOS ONE Editorial staff should be able to do this on my behalf. They are very minor edits and can be completed in 15-20 minutes by the authors.

7. PLOS authors have the option to publish the peer review history of their article (what does this mean?). If published, this will include your full peer review and any attached files.

Reviewer #1: **Yes: **Sharon L Kendall

Reviewer #2: **Yes: **David E MacHugh

---

## [Author Response · Author response to Decision Letter 1]

8 Sep 2020

Dear PLOS ONE Editor, 

We thank you for your review of the manuscript entitled “Severity of bovine tuberculosis is associated with innate immune-biased transcriptional signatures of whole blood in early weeks after experimental Mycobacterium bovis infection.” We have addressed the second round of comments suggested by the reviewer and hope that we have taken care of all concerns. 

Reviewer comments

The new Figure 1 is an excellent graphical representation of the work and make it much easier for readers to understand how the study was conducted.

We hoped this figure would improve the manuscript and thank the reviewer for their comment. 

Line 54-55: This statement doesn't currently make any sense. The citation is from 1995 and the year being referred to 2004. Is there a citation for the $50 million cattle statistic? If so, the text could be re-worded as follows - note insertion of new citation and "at least" before "$3 billion"

"In 2004, it was estimated that worldwide, an estimated >50 million cattle are infected with M. bovis (CITATION), resulting in a financial loss of at least $3 billion USD annually (6)."

We apologize for the confusion with this statement. The “2004” date is a mistake, so that has been corrected. We have not been able to find a more recent citation for the number of cattle and cost associated with bovine tuberculosis worldwide. While old, this is the citation that is still currently used by ourselves and others. 

Lines 258 and 259: Please make it clear you are using Log2FC values (with subscript "2"). "LogFC" is not sufficient - a non-expert reader will assume this is Log10.

Lines 275 and 276: Same as previous comment.

We apologize for the oversight, this has been addressed.

Supplementary Files 3 and 4: The terms "upregulated" and "downregulated" are still being used inappropriately in these files. Please change to "increased expression" or "decreased expression" etc. as detailed in the review of the first version of the manuscript.

We apologize for the oversight, we have made these changes accordingly. 

Again, we thank the editor and reviewers for their time and effort in reviewing our manuscript. We hope that we have addressed all concerns and hope that you will find this manuscript suitable for publication. 

Thank you, 

Paola M. Boggiatto

---

## [Editor Report · Decision Letter 2]

16 Sep 2020

Severity of bovine tuberculosis is associated with innate immune-biased transcriptional signatures of whole blood in early weeks after experimental Mycobacterium bovis infection

PONE-D-20-09070R2

Dear Dr. Boggiatto,

We’re pleased to inform you that your manuscript has been judged scientifically suitable for publication and will be formally accepted for publication once it meets all outstanding technical requirements.

Kind regards,

Ann Rawkins, PhD

Academic Editor

PLOS ONE
---

## [Editor Report · Acceptance letter]

22 Sep 2020

PONE-D-20-09070R2 

Severity of bovine tuberculosis is associated with innate immune-biased transcriptional signatures of whole blood in early weeks after experimental *Mycobacterium bovis* infection 

Dear Dr. Boggiatto:

I'm pleased to inform you that your manuscript has been deemed suitable for publication in PLOS ONE. Congratulations! Your manuscript is now with our production department. 

Kind regards, 

on behalf of

Dr. Ann Rawkins 

Academic Editor

PLOS ONE